# DevFly: Bio-inspired Development of Binary Connections for Locality Preserving Sparse Codes

**Tianqi Wei** [*]
School of Informatics
University of Edinburgh
Edinburgh, UK EH8 9AB
tianqi-wei@outlook.com

**Rana Alkhoury Maroun**
School of Informatics
University of Edinburgh
Edinburgh, UK EH8 9AB
rana.e.elkhoury@gmail.com

**Qinghai Guo**
ACS Lab
Huawei Technologies
Shenzhen, China
guoqinghai@huawei.com

**Barbara Webb**
School of Informatics
University of Edinburgh
Edinburgh, UK EH8 9AB
B.Webb@ed.ac.uk

## Abstract

Neural circuits undergo developmental processes which can be influenced by experience. Here we explore a bio-inspired development process to form the connections in a network used for locality sensitive hashing. The network is a simplified model of the insect mushroom body, which has sparse connections from the input layer to a second layer of higher dimension, forming a sparse code. In previous versions of this model, connectivity between the layers is random. We investigate whether the performance of the hash, evaluated in nearest neighbour query tasks, can be improved by process of developing the connections, in which the strongest input dimensions in successive samples are wired to each successive coding dimension. Experiments show that the accuracy of searching for nearest neighbours is improved, although performance is dependent on the parameter values and datasets used. Our approach is also much faster than alternative methods that have been proposed for training the connections in this model. Importantly, the development process does not impact connections built at an earlier stage, which should provide stable coding results for simultaneous learning in a downstream network.

## 1 Introduction

The insect mushroom body (MB) is a learning centre in the insect brain, and its function is of interest as a neural architecture for efficient association of arbitrary sensory patterns to actions. The MB is a shallow network, with an input layer of projection neurons (PNs) that sparsely connect to a larger number of Kenyon cells (KCs), which are fully connected to a small number of MB output neurons (MBONs). In this paper we focus on the PN to KC connectivity, and consider how the activity pattern in KCs could provide an efficient code for the input that enhances the appropriate generalisation from previously experienced patterns to new inputs. In the MB, only a small portion of KCs is activated at any time (Honegger et al., 2011), which is known as sparse coding (Olshausen & Field, 2004). Sparse coding facilitates learning by letting each KC represent a more specific pattern of sensory inputs, which reduces unintended association with a different pattern during learning (Treves & Rolls, 1991). Ideally, similarity in sensory input space should map to similarity in the space of sparse codes.

---

[*]Now at School of Artificial intelligence, Sun Yat-sen University, Zhuhai, China.

36th Conference on Neural Information Processing Systems (NeurIPS 2022).

One way to interpret the function of the mapping from PN to KC is treating the mapping as a locality-sensitive hash (Dasgupta et al., 2017). Ideally, a locality-sensitive hash (LSH) function can preserve relative distances between data samples in the corresponding hash coding results (Hutchins, 1999), such that the codes can be used to efficiently locate close samples. Compared to a typical LSH, the MB-inspired hash algorithm (Figure 1) proposed by Dasgupta et al. (2017), called FlyLSH, maps inputs (with a dimension $d$) to a higher dimensional space (with a dimension $m$), using sparse binary connections, and then creates a sparse binary code by choosing the top $k$-values and setting them to 1. FlyLSH achieved better performance than a standard LSH algorithm with the same hash length [2]. This in turn allows FlyLSH to attain higher precision than LSH algorithms that use a dense connectivity matrix. It has inspired some variations including deep nets that use FlyLSH as initial layers. For example, work by Chancan et al. (2020) combined a FlyLSH based network with a one-dimensional continuous attractor neural network and achieved a better performance than other methods in visual place recognition tasks.

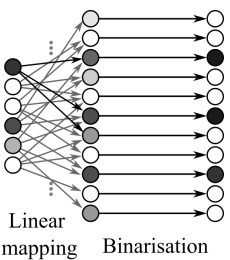

Figure 1: Two steps to produce a hash using binary connections.

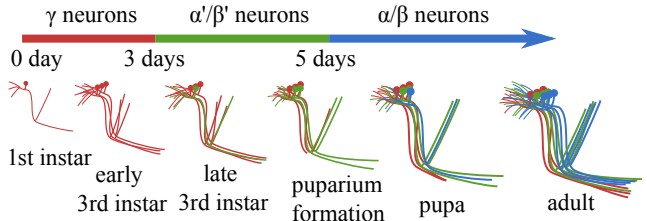

Figure 2: Summary of the mushroom body development (modified from (Lee et al., 1999)).

FlyLSH uses fixed random connections between the PN and KC layers, a feature inspired by previous accounts of this connectivity in the MB (Caron et al., 2013; Masuda-Nakagawa et al., 2005). However, more recent biological results suggest that the formation of connections between PNs and KCs can be experience-dependent, and when a KC connects to PNs depends on (1) when the KC is developmentally generated and (2) the existing connections (Eichler et al., 2017). Like other neural circuits, the MB develops during the animal's maturation from a few neurons to thousands of neurons, and new connections are built during this process. In the fly, this happens during embryo, larvae, pupae, and early adult stages (Figure 2) (Lee et al., 1999), with and without external stimulus. There are also evidences suggesting that the distribution of the connections cannot be simply explained as random (Eichler et al., 2017; Elkahlah et al., 2020; Hayashi et al., 2021).

In this paper, we propose a bio-inspired method that develops the binary connections between input PNs and the coding KCs. The connections to each coding dimension are built according to strongest dimensions of a selected sample, and once built, they will not be impacted by later development, so that reliable coding results are preserved during the development process. We then test this method against FlyLSH in a nearest neighbours search task using several datasets as inputs, to investigate whether the performance can be improved by an MB-inspired process of developing the connections.

## 2 Related works

Models of the MB have been presented by several different groups mostly in the context of associative learning or classification tasks, such as (Huerta et al., 2004; Smith et al., 2008; Mosqueiro & Huerta, 2014; Bennett et al., 2021; Wessnitzer et al., 2012; Ardin et al., 2016; Zhu et al., 2021). Most of these studies make the assumption that the connections between PNs and KCs are randomised. The effect of non-random PN to KC connections has been explored for the biological task of odour discrimination in a model by Zavitz et al. (2021). They find the effective dimensionality of the KC layer is reduced, the response to odours potentially of ethological significance to the fly are heightened and the generalisation between odours is improved. There are also recent models that

---

[2]Note there is a difference in the definition of the hash length of FlyLSH and that of a standard LSH. The hash length of FlyLSH is counted by the number of KCs that activate, but the hash length of a standard LSH is counted by both activated and non-activated coding dimensions.

include plasticity between PNs and KCs, such as MothNet (Delahunt & Kutz, 2019) and a model by Peng & Chittka (2017) which adjusts the PN-KC synapses according to reward modulation and coactivation of connected PN and KC, showing that this alters the generalisation/discrimination tradeoff for KC encoding of sensory patterns.

A number of papers have explored the effects of altering the connection scheme from PN to KC from the FlyLSH default, which is for each KC to receive inputs from $n$ randomly selected PNs where $n$ is some fraction of the input dimension. Xu & Qiao (2018) explored the use of a variable $n$, centred on the same mean, and reported improved performance over FlyLSH for nearest neighbour search in CIFAR-10, as well as better performance than several data dependent (but unsupervised) standard hash methods including PCA-hash, Spherical Hash and ITQ. Data-dependent methods for forming the sparse connections should be expected to do better than random connections for the same $n$. Ma et al. (2020) proposed an algorithm to iteratively construct the projection matrix to produce the winner-take-all (WTA) output that maximises the distance between codes for the whole dataset, subject to the limit $n$ on the number of connections per KC. Ryali et al. (2020) proposed a method similar to self-organising mapping to train a modified version of FlyLSH, which uses continuous values instead of binary values for mapping, noting that this is equivalent to k-means clustering with some additional constraints. Both methods outperform FlyLSH, but are relatively data greedy because they use all images in the datasets for learning the connections. In Pehlevan et al. (2017) a neurally plausible method for online k-means clustering is proposed and related to known PN-KC connectivity properties, although the method is not explicitly tested as a hashing code.

Preissner & Herbelot (2019) took a contrasting approach when applying the FlyLSH method to a language corpus. In their work the PN layer is incrementally expanded, and the KC connections randomly redistributed, while preserving equal $n$, as words are encountered in new contexts. An interesting point made in this approach is that the resulting sparse KC code is interpretable, for example, the set of PNs connecting to one KC tend to refer to related or frequently co-occuring words in the input data. However, like the previous methods, it requires multiple updates for every KC during the training process.

There is relevant earlier research on growing ANNs (also described as resource allocation networks (Platt, 1991)) where neurons and/or connections are gradually added to the network (see review by Macleod & Maxwell (2001) and more recently (Mixter & Akoglu, 2020)). In this approach, new units are added when the existing network does not meet some criteria with respect to newly presented data, and then all the connection weights are retrained using gradient descent or similar standard methods. These methods might also include pruning, to remove units that are not contributing to successful performance, e.g., (Yingwei et al., 1998). In either case the aim is to produce a more compact network size. By contrast, our developmental algorithm assumes a target total number of neurons, and tries to connect each successive one to represent the data efficiently with respect to previous connections, but without changing any existing connections. Interestingly, our approach (and the MB structure) can be more closely compared to early work on 'prototype' learning in the form of RCE networks (Reilly et al., 1982), where input samples are used to set the weights of successive nodes in the first layer, with new connections to the next node formed if the existing node activations cannot support successful classification in the following layer, but no further changes made to existing weights. RCE networks have also been used in an unsupervised form (Marsland et al., 2002) where nodes are added when none of the current nodes are sufficiently activated.

## 3 Methods

The aim of the algorithm presented here is to produce a locality sensitive hash code such that the distance in the sample space is positively correlated to the distance in the code space. In the following, the distance in sample space is measured as Euclidean distance, and the distance in the code space is measured as Hamming distance for the binary coding (Sharma & Navlakha, 2018).

Hashing by FlyLSH or its variations consists of two steps (Figure 1), a linear mapping from the input (with dimension $d$) to the code space (with dimension $m$), and binarisation of the resulting activity pattern. Note that following FlyLSH, we equalise the mean of the pixel intensities across images (Dasgupta et al., 2017) by substracting the mean from each sample so that the sum of all the pixels is zero.

In FlyLSH, the linear mapping step uses a binary matrix $R$ generated by randomly choosing $n = \lfloor \alpha d \rfloor$ indexes from the input dimensions for each code dimension, where $0 < \alpha < 1$, typically $\alpha \sim 0.1$, and $\lfloor \cdot \rfloor$ is the floor operation. Thus, for each row of $R$, $n$ elements are randomly set to 1, and the remainder are 0. After all of the coding dimensions are computed, they are binarised by choosing the top $k$ values and setting them to 1, with all remaining values set to 0.

Note that although in the fly, and in contrast to standard LSH, $m \gg d$, in practice FlyLSH is often tested with high dimensional inputs resulting in $m \leq d$. The key properties preserved from the fly are the use of a low sampling ratio $\alpha$ rather than full connectivity, and a sparse code $k \ll m$. For clarity, we have not followed Dasgupta et al. (2017) in referring to $k$ as the hash length, as in practice the full code of length $m$ is used to calculate Hamming distance, but see section 4.1 below.

The method we propose (which we call DevFly) uses bioinspired development processes to replace the random choice that generates matrix $R$. It mimics the biological observation that in the MB, individual KCs mature at different times and each will make connections to PNs as they mature, that is, there is no single stage at which all PN-KC wiring occurs. We assume that, in the MB, when some criteria is triggered, synapses are built from those PNs that have strongest activities to an immature KC; subsequently the connections for this now mature KC remain fixed. Correspondingly, in DevFly, when some criteria is triggered, the connections are built from strongest input dimensions to a new code dimension. We propose three alternative criteria, which are referred to as Method 1,2 and 3. Method 1 is shown in Algorithm 1. Method 2 and 3 are shown in Algorithm 2. The code is available in the supplementary materials.

**Algorithm 1** Connection development for Method 1

**Input**: Data Samples $S = \{\mathbf{s}_1, \mathbf{s}_2, \ldots \mathbf{s}_q, \ldots \mathbf{s}_u\}$.
**Parameter**: code dimension $m$, number of connections per KC $n$.
**Output**: KC weights $R = \{\mathbf{r}_1, \mathbf{r}_2, \ldots \mathbf{r}_j, \ldots \mathbf{r}_m\}$.

1: $j := 1$
2: # Randomly choose $m$ samples from $S$.
3: **for** $\mathbf{s}_q$ in $m$ samples from $S$ **do**
4:     # Set the weights of a KC according to a sample.
5:     Find the indexes $\mathbf{i}' \in \mathbb{Z}^n$ of top $n$ values in the sample.
6:     $r_{i' \in \mathbf{i}'} := 1$, where $r_i \in \mathbf{r}_j$.
7:     j++
8: **end for**
9: **return** $R$

**Algorithm 2** Connection development for Method 2 and 3

**Input**: Data Samples $S = \{\mathbf{s}_1, \mathbf{s}_2, \ldots \mathbf{s}_q, \ldots \mathbf{s}_u\}$.
**Parameter**: threshold $\theta$, threshold increasing rate $\beta$, code dimension $m$, number of connections per KC $n$.
**Output**: KC weights $R = \{\mathbf{r}_1, \mathbf{r}_2, \ldots \mathbf{r}_j, \ldots \mathbf{r}_m\}$.

1: Let $R := \mathbf{0}$, $j := 1$, $q := 1$.
2: **while** $j \leq m$ **do**
3:     # Iterate for next unfamiliar sample $\mathbf{s}_q$.
4:     **while** $\exists\, j' \in \{1, 2, \ldots, j\}$ such that $\mathbf{r}_{j'} \mathbf{s}_q > \theta$ **do**
5:         $q + +$
6:         **if** $q > u$ **then**
7:             $\theta := \beta \theta$
8:             $q := 0$
9:         **end if**
10:    **end while**
11:    # Set the weights of a KC according to a sample.
12:    **if** Method 2 **then**
13:        Find the indexes $\mathbf{i}'$ of top $n$ values in the sample.
14:    **else if** Method 3 **then**
15:        Randomly choose $n$ indexes $\mathbf{i}'$ using the normalised sample as probability.
16:    **end if**
17:    $r_{i \in \mathbf{i}'} := 1$, where $r_i \in \mathbf{r}_j$.
18:    $j + +$.
19: **end while**
20: **return** $R$

**Method 1: connection development according to randomly picked samples.** The first method is designed to test the importance of connecting from the strongest input dimensions, by simply setting the criteria to be randomness: (1) Randomly choose data samples, as many as the number of code dimensions. (2) Given a sample, sort the sample dimensions by their input value and take the $n$ largest. (3) Connect the largest $n$ input dimensions to a code dimension. (4) Repeat the previous step until all code dimensions are connected.

**Method 2: connection development according to unfamiliar samples** In the second method, we set the criteria so as to build new connections only when a sample input is not already well connected.

That is, given a sample, an unconnected code dimension will be connected to the strongest input connections only if none of the previously connected code dimensions have an output stronger than a threshold. The threshold can be interpreted as a means of recognising "familiar" samples. (1) Set a threshold for when to connect. (2) Given the first sample, sort the sample dimensions by their input value and take the $n$ largest. (3) Connect the largest $n$ input dimensions to a code dimension. (4) For the next sample, check the response of connected code dimensions. If all them are weaker than the threshold, connect an unconnected code dimension according to the large $n$ input dimensions in this sample. (5) Repeat the previous step until all code dimensions are connected. If all samples have been examined but some code dimensions remain unconnected, increase the threshold, for example, by timing a factor $\beta$, and then loop through the samples again.

**Method 3: connection development randomly according to unfamiliar samples** We propose an additional variation on Method 2 by introducing noise when building the connections. Instead of connecting the input dimensions and code dimensions strictly according to the strongest input dimensions, the connection is built by randomly selecting input dimensions according to the distribution of the input dimensions. This seems biologically more realistic, but we also believe that with this modification, the new connections could have better generalisation while capturing the features.

The code is available on GitHub [3].

# 4 Experiments

We applied the three connection development methods to FlyLSH, and tested them on four different datasets for the precision in finding nearest neighbours. The datasets are MNIST (CC BY-SA 3.0), CIFAR-10 (MIT), SIFT10M and GloVe (Apache-2.0). MNIST (LeCun et al., 1998) is a dataset with images of handwriting digits from 0 to 9. Each of the images has a size of $28 \times 28$ and is in greyscale. CIFAR-10 (Krizhevsky, 2009) is a dataset with images from ten classes of objects. Each of the images has a size of $32 \times 32$ for each RGB channel. SIFT10M (Jegou et al., 2010) also contains vectors representing images but has a smaller dimension (128) and is specifically designed for nearest neighbours search. The GloVe dataset (Pennington et al., 2014) contains pre-trained vectors with a dimension of 300 for word representation, and the vectors have linear substructures in their space. These datasets do not include personally identifiable information or offensive content.

Following the protocol in the experiments by Dasgupta et al. (2017) and Sharma & Navlakha (2018), for each of the datasets, we chose 10000 samples as an experiment dataset for the nearest neighbours search. In each experiment, $m$ samples from a dataset were used to train the model, then 1000 samples were randomly chosen for querying. For each query sample, we listed the 200 samples with the smallest Hamming distances to the query sample, then compared the list with the true neighbours list found according to their Euclidean distances. We report the results for the three DevFly methods with different hyperparameters, varying $k$, $m$ and $\alpha$. Each of the tests with a specific combination of hyperparameters was repeated 10 times for statistical results. For comparison, we also run the random FlyLSH described by Dasgupta et al. (2017) on the same datasets. Because the original code by Dasgupta et al. (2017) is not available, we used code published by the same research group (Sharma & Navlakha, 2018) to implement these methods. We noticed that the performances reported for FlyLSH are not consistent in those two papers. Hence, comparisons are based on our experimental results only. For reference, both Fly methods consistently outperform a traditional LSH (Hutchins, 1999) if its hash length is set to $k$.

## 4.1 Query time

An LSH using a sparse binary code based on winner-take-all (WTA) has an advantage in query time. Because the number of activating coding dimensions is fixed, to carry out a nearest neighbour search query for a given sample it is not necessary to compare all of the coding dimensions. Instead, it is sufficient to just compare those coding dimensions which are activated (coded as 1) by the query sample. Hence, the query time will scale with $k$ rather than with $m$. This provides a potential advantage over DenseFly (Sharma & Navlakha, 2018). DenseFly varies from FlyLSH by not using top-$k$ WTA but instead using a threshold to set code elements to $1$ or $0$. As we show in figure 3 the mean average query time for Method 1 using a fixed sparseness, i.e., $k = m/20$ is significantly lower.

---

[3]`https://github.com/InsectRobotics/DevFlyPublication.git`.

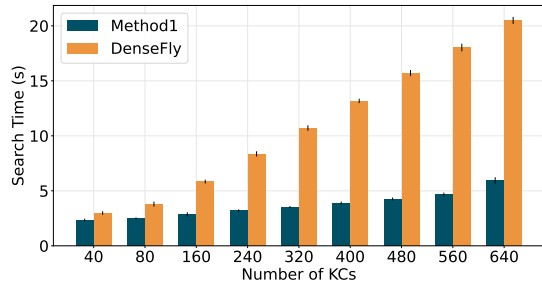

Figure 3: The search speed advantage of using a sparse code (Method 1) instead of a dense code (DenseFly). Tested on MNIST with 10000 samples. The search time is the time taken for 1000 queries. They were repeated for 10 times to compute the average and variance. Each query retrieves 200 nearest neighbours. Hash codes of all the samples, including the query samples, were computed before the queries; for Method 1, the sparse coding uses a fixed $k = m/20$. Please note the 10 repeats were executed at the same time in parallel to simulate concurrent queries, so a much shorter time is expected if the repeats are sequential. Tested on a computer running Ubuntu 20.04 with Intel®Core™i9-10940X CPU with 28 hyperthreading logical cores. The code was implemented in Python and most of the code for testing these two methods were shared.

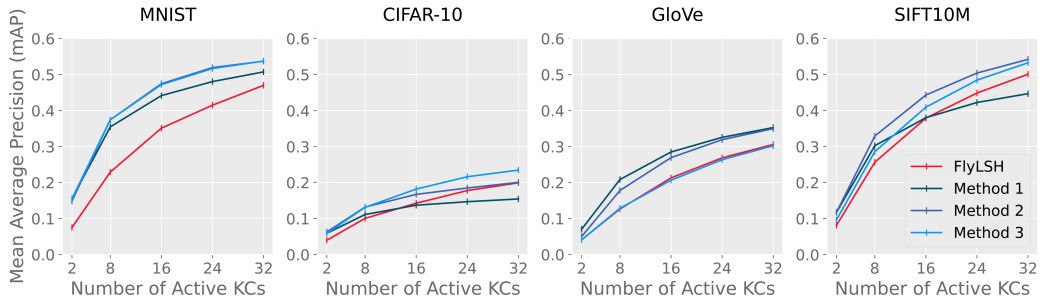

Figure 4: Performance on four datasets. 10000 samples were taken from each dataset for the tests. 1000 queries for 200 nearest neighbours were made for the means and variances. $m = 20k$, $\alpha = 0.1$.

The same advantage applies to our other developmental methods (and to the original FlyLSH) as they also use top-$k$ WTA for binarisation.

## 4.2 Search accuracy

DevFly methods are tested on four datasets and the mean Average Precision (mAP) for nearest neighbour query is shown is Figure 4. Precision here measures the ratio of true neighbours in the neighbours found by our methods. The average precision is the precision of 1000 queries in an experiment, and the mAP is the mean of average precision for ten experiments. In these experiments, $\alpha = 0.1$, sparseness $m/k = 20$, but $k$ varies. We also tested the effects of changing the sparseness by varying $m/k$. The results are shown in supplementary Figures S.1 to S.4.

All the developmental methods show a similar performance on MNIST, and perform better than FlyLSH for all $k$. For MNIST, the relative improvements range from 9% for $k = 32$ to 106% for $k = 2$, even with Method 1, in which the connections are made without any criteria for choosing the samples used to determine the connections.

For the other datasets, at least one developmental method outperforms FlyLSH, although which method performs best varies with the dataset. Notably for GloVe, the randomness introduced in Method 3 produces no improvement on performance over FlyLSH. For CIFAR-10, both the improvement with DevFly methods and the overall performance are relatively low. This might be caused by the different complexity of this dataset compared to MNIST. CIFAR-10 has larger

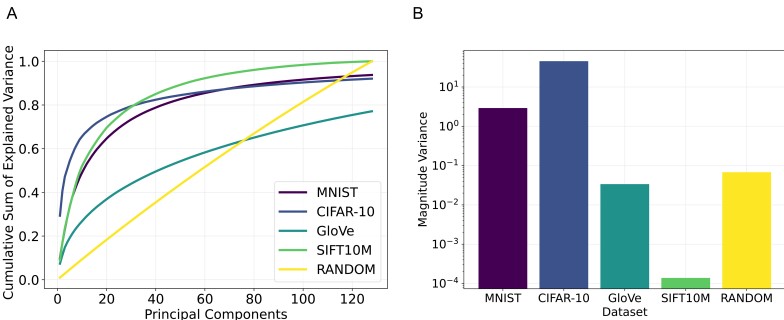

Figure 5: (A) Cumulative sum of explained variances of the principal components of the four datasets, plus a randomly generated dataset with a uniform distribution. (B) Variances of the sample dimensions across the same datasets. The y-axis is plotted in log scale.

dimension, $d = 32 \times 32 \times 3 = 3072$, than MNIST, $d = 28 \times 28 = 784$. More crucially, for MNIST, the information for distinguishing the digits is clearly given by the highest intensity pixels, so our method which forms connections based on intensity will correspond to the key distinguishing features of the data. For illustration, we can visualise the weights for 100 KCs formed by Method 2 after 320 steps of development which clearly reflect the structure of the data (Figure S.6). Images in CIFAR-10 are photos of real objects, which have richer and more complex details than images in MNIST, and the pixels with the strongest intensities are not necessarily the most informative, e.g. they might be part of the background, corresponding to sky.

The higher performance of the models on MNIST and SIFT10M than on GloVe is, we believe, due to the data in MNIST and SIFT10M lying on a lower dimensional manifold, relative to their input dimensions, than GloVE, and hence requiring fewer samples to adequately represent the space. Principal components of the variables of a dataset can give us a good indication of the size of this space, and as shown in Figure 5 (A), the cumulative sum of explained variance by principle components increases much more slowly for GloVe than the other datasets. This is confirmed by observing that none of the models was able to find more than 10% of the nearest neighbours when tested on a randomly generated dataset (see section S.B.4). However, this does not explain the worse performance on CIFAR-10. Instead, we observe that, ideally, to preserve the nearest neighbours in the input space, each KC (which represents a centroid in space) should be activated proportionally to the Euclidean distance between the input sample and the centroid. However, as the intermediate code by KCs $\mathbf{y}'_j := R\mathbf{s}_q$ is actually defined by the dot product, the activations are biased by samples that have a bigger norm. This is not the case with datasets that have samples with equal L2 norms, as the dot product is local when samples lie on the same hypersphere (SIFT10M) or have approximately this property (MNIST). Experimentally, we can test this by checking if equalising the norms improves the performance, and we indeed see a significant improvement for CIFAR-10 (Figure S.8). However, if the goal is LSH, equalising the norms should not be considered to be an acceptable pre-processing step, as it does not preserve the distances of the original dataset.

### 4.3 Learning efficiency

There are many existing methods for tuning the projection matrix in LSH, and the methods can be categorised into three groups, unsupervised, supervised, and semisupervised. These methods can also be further categorised according to the levels of supervised information, linearity, one-shot/multiple-shot. For a comprehensive review, see the survey by Wang et al. (2016). Among them, we have not found a development based method.

Another alternative approach to optimising PN-KC connections for an MB-inspired hashing model is BioHash (Ryali et al., 2020). BioHash proposed a method similar to self-organising mapping to train a modified version of FlyLSH, which uses a dense connection matrix and continuous values instead of binary values for mapping. We modified BioHash in order to have the same number of active KCs as DevFly and FlyLSH. We find that BioHash takes much more time to reach the same precision level as either Method 1 or Method 2 (Figure 6). Only 320 iterations were needed for DevFly versus 10000 iterations for BioHash to reach the same performance level. In application, fast convergence

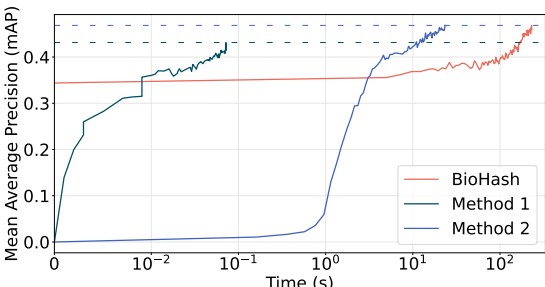

Figure 6: Performance over training time for Method 1 and Method 2 versus BioHash. Note that the scale of the x-axis is semilog; it is linear from 0 to $10^{-2}$ and logarithmic for the rest of the range.

of the model helps for timeliness. For huge multi-dimensional datasets, the bigger the dataset, the stricter the computational requirements are, and thus DevFly can be used in cases where BioHash is too costly. This also means that DevFly is more suitable for cases where the available training data is scarce.

### 4.4 Insensitivity to sample order

We also tested DevFly with ordered samples to check if training with non-random samples would impact performance. Because the original MNIST dataset is in a random order, we sorted 10000 samples according to their labels. Then, in Method 1, $m$ samples are randomly chosen from the ordered data and used to form the successive connections to KCs while maintaining their labelling order. For Methods 2 and 3, in training, all samples labelled 0 were fed to the model, then all samples labelled 1 were fed, etc. Methods 2 and 3 may loop all samples several times before all KCs are connected if the initial threshold is small and $\beta$ is close to 1, but the samples keep ordered. The results suggest that for all three methods, the order does not impact the performance (Figure S.5). Hence, training with a development-like approach, which only builds new connections and keeps the existing connections, helps to reduce the sensitivity to the order of samples. Note that FlyLSH results are not impacted by the order of samples simply because it does not use the dataset at all in initialising the matrix $R$.

## 5 Discussion

In this paper, we proposed a method to improve the performance of FlyLSH by developmental initialisation of its binary connections from input dimensions to code dimensions. This was inspired by biological data showing that in the insect mushroom body (the neural circuit that inspired FlyLSH) the connections are not random but there is a process of development in which PNs connect to KCs. We implement this as a simple wiring process driven by the highest values in data samples. We show this improves the performance of FlyLSH in a range of nearest neighbour search tasks. Essentially, FlyLSH detects features of the input space and represents them in a sparse high dimensional code. Using random connections can result in detectors for features that do not exist in the dataset, whereas DevFly constructs the feature detectors from samples in the dataset, hence improves the efficiency of the coding for non-uniform dataset. The advantage provided by DevFly diminish as the feature space gets larger (Table S.2, S.3, and S.4) as the random method covers more of the space.

A limitation of our methods that this process is not guaranteed to produce the optimal solution, but as we have shown it can nevertheless be effective for LSH problems. There are many known methods for optimising the (sparse) connectivity between layers to preserve locality (or other desired) properties of the data. However most of these require multiple passes through a large number of randomly ordered samples from the dataset and are significantly more costly to compute. The method we use here is aimed at scenarios - such as real brains - where 'rewiring' is costly, and a method that requires all weights to be adjusted for each new sample is expensive, as well as potentially disruptive as each new input changes the existing encoding.

The improvement in performance of DevFly over FlyLSH is due to an overlap between the regions with highest variance and those with highest intensity (Figure S.7). Essentially, our method assumes that the strongest signals in the input dimensions correspond to the most relevant signals for similarity (here defined as Euclidean distance in the input data). In CIFAR-10, the dimensions containing the highest intensity pixels don't overlap with the dimensions that vary the most (Figure S.7). By relying on the highest pixel activations to produce the hash, the most important dimensions in CIFAR-10 will be overlooked, leading to poor results. For the other datasets, where high intensity pixels coincide with high variance, DevFly has an advantage because it disregards those areas with low variance, whereas FlyLSH sampling randomly will weight them equally. Hence a possible way to improve performance in this case would be to pre-process the data to extract the features relevant to similarity - in a sense this is what we see already for SIFT10M.

Note that in some cases, hashing methods include additional steps to increase performance. For example, multi-bin search by Kamel et al. (2015), a two-step evaluation is used to find the nearest neighbours: (1) the hash is used to find the bins that contain similar samples, (2) nearest neighbours are determined according to other information, which can be the raw data. The second step can significantly improve the accuracy. Other methods compute several hash tables and use them to compare across different results for the same query. The models used here only focus on the first step, and were trained only by learning some samples once, so the precision is expected to be lower than state-of-the-art hashing. Adding additional steps is still feasible to further improve the performance.

Our results also show this method is better suited to datasets that are clustered, have samples with a relatively constant norm, and in general have higher values in those input dimensions that are relevant to the similarity measure. For consistency with previous work we have used Euclidean distance in the input space (after image intensity equalisation) as the 'ground truth' for similarity, but note this may not be the best measure for datasets such as CIFAR-10. Of interest is whether our method could be adapted to develop wiring from the input dimensions that show most correlated variability to the code dimensions. For example, wiring from the changing pixels of a security camera video.

The fact that a weight in DevFly methods only updates once for the entire life of the model facilitates the implementation of this model to hardware. For example, programmable ROM, which permits data to be written to memory only once, can be used to save the weights while learning. It is also easy to be implemented on an integrated circuit with minimum die size usage. The implementation of Method 3 on hardware does not necessarily need a random number generator. In fact, Method 3 suggests that for datasets like CIFAR-10, variation that naturally results from unreliable hardware can be an advantage for better generalisation, providing higher fault tolerance for hardware.

In the insect MB and the mammal cerebellum, sparse coding is used to extract and separate low-level features across different neurons (Olshausen & Field, 2004; Farris, 2011). Hence, DevFly should also be suitable to improve the detection of low-level features in shallow layers of a deep net, such as the deep net used by Chancan et al. (2020) for place recognition. Error propagation-based approaches have difficulties to train shallow layers due to averaging of the gradient during propagation. Although there are some tricks like ResNet to convey more information by shortcut connections, it is worth considering the use of an unsupervised approach to update shallower layers directly according to data samples. Importantly, a developmental training approach provides more stable signals for deeper layers during training as it does not alter existing connections.

A particular application of this model of interest to us in future is reinforcement learning (RL). An approach to improve exploration efficiency in RL is to count visited states for computing the novelty. For continuous tasks or tasks with high-dimensional observations, using the raw observation as a state to count is not feasible. Tang et al. (2017) proposed a model that uses LSHs instead of the raw observation for state counting which achieved "surprisingly good results". They claimed that static hash methods such as SimHash perform worse than methods that can learn from data, such as autoencoders. However, they also argued that it is important to keep the mapping from states to codes relatively consistent over time, otherwise the same state could be mapped to different codes by training. DevFly suits this application because each weight only updates once so the learned coding dimension never changes, i.e., improving coding quality while minimising unnecessary change.

LSH nearest neighbour classification can be used for a wide range of applications from security camera recognition to medical image analysis. By introducing a biologically inspired method we might hope that the resulting system would more closely resemble the decisions made by humans and hence have a positive societal impact. We also note that the simplicity of the approach makes

it suitable for edge computing, which could allow data (such as that from security cameras) to be processed locally, with consequence privacy advantages in limiting data transfer. As this work emphasises the development of connections and can belong to a special case of optimisation, it does not have a direct negative societal impact.

## Acknowledgement

Funding: This work was supported by the Huawei Technologies Co.,Ltd. [grant number YBN2020045132]

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
