# Supplementary for "DevFly: Bio-Inspired Development of Binary Connections for Locality Preserving Sparse Codes"

**Tianqi Wei** [*]
School of Informatics
University of Edinburgh
Edinburgh, UK EH8 9AB
tianqi-wei@outlook.com

**Rana Alkhoury Maroun**
School of Informatics
University of Edinburgh
Edinburgh, UK EH8 9AB
rana.e.elkhoury@gmail.com

**Qinghai Guo**
ACS Lab
Huawei Technologies
Shenzhen, China
guoqinghai@huawei.com

**Barbara Webb**
School of Informatics
University of Edinburgh
Edinburgh, UK EH8 9AB
B.Webb@ed.ac.uk

## S.A    Methods

| | Description | Definition or Typical value |
|---|---|---|
| $S$ | Data Samples | $S = \{\mathbf{s_1}, \ldots \mathbf{s}_q, \ldots \mathbf{s}_u\}$ |
| $q$ | Sample index | |
| $u$ | Number of samples | |
| $\mathbf{s}$ | A data sample | $\mathbf{s} = \{x_1, \ldots x_i, \ldots x_d\}$ |
| $d$ | Input dimension | |
| $i$ | Input dimension index | |
| $\alpha$ | Sampling ratio | $0 < \alpha < 1, \alpha \sim 0.1$ |
| $n$ | Number of connections per KC | $n = \lfloor \alpha d \rfloor$ |
| $k$ | Number of active KCs | |
| $m$ | Code dimension (hash length) | |
| $j$ | Code dimension index | |
| $k/m$ | Sparseness | |
| $\theta$ | Threshold | $\theta \sim 10$ |
| $\beta$ | Threshold increasing rate | $1 < \beta, \beta \sim 1.2$ |
| $\mathbf{y}'$ | Intermediate code | $\mathbf{y}' = \{y'_1, \ldots, y'_j, \ldots, y'_m\}$ |
| $\mathbf{y}$ | Hash code | $\mathbf{y} = \{y_1, \ldots, y_j, \ldots, y_m\}$ |
| $R$ | Mapping matrix | $R = \{\mathbf{r}_1, \ldots, \mathbf{r}_j, \ldots, \mathbf{r}_m\}$ |
| $\mathbf{r}$ | Connections to a code dimension | $\mathbf{r} = \{r_1, \ldots, r_i, \ldots, r_d\}, r_i \in [0, 1]$ |
| $\gamma$ | expansion ratio | $\gamma = m/d$ |

Table S.1: Notations in this paper

[*]Now at School of Artificial intelligence, Sun Yat-sen University, Zhuhai, China.

36th Conference on Neural Information Processing Systems (NeurIPS 2022).

## S.B Experiments

### S.B.1 Search accuracy

| $k$ | FlyLSH | Method 1 | Method 2 | Method 3 |
|---|---|---|---|---|
| 2 | 20.65(0.98) | 38.06(0.60) | 36.60(0.29) | 36.44(0.74) |
| 4 | 26.31(1.32) | 48.69(0.46) | 45.22(0.44) | 46.00(0.48) |
| 8 | 34.79(1.38) | 56.53(0.42) | 55.18(0.24) | 55.37(0.47) |
| 12 | 41.33(0.94) | 59.93(0.50) | 58.65(0.25) | 59.29(0.30) |
| 16 | 45.41(0.81) | 61.72(0.44) | 61.40(0.35) | 61.34(0.38) |
| 20 | 48.10(0.95) | 62.91(0.37) | 62.94(0.53) | 62.77(0.44) |
| 24 | 49.29(0.99) | 63.92(0.38) | 64.09(0.25) | 63.75(0.47) |
| 28 | 51.65(1.17) | 64.42(0.42) | 64.44(0.25) | 64.70(0.40) |
| 32 | 53.35(0.76) | 65.20(0.29) | 64.66(0.20) | 65.30(0.28) |

Table S.2: Mean average precision of nearest neighbours search on 1000 random ordered samples in MNIST. The numbers are in percentage and the numbers in brackets are variances.

| $k$ | FlyLSH | Method 1 | Method 2 | Method 3 |
|---|---|---|---|---|
| 2 | 7.49(0.46) | 15.89(0.60) | 16.88(0.40) | 15.97(0.47) |
| 4 | 12.96(0.77) | 25.17(0.64) | 25.94(0.35) | 26.22(0.56) |
| 8 | 22.92(0.95) | 35.61(0.70) | 37.68(0.41) | 37.62(0.61) |
| 12 | 30.38(0.48) | 41.34(0.82) | 43.56(0.42) | 43.25(0.58) |
| 16 | 35.13(0.58) | 44.48(0.61) | 46.89(0.33) | 47.10(0.60) |
| 20 | 39.21(0.69) | 46.68(0.51) | 49.66(0.26) | 49.19(0.50) |
| 24 | 41.90(0.72) | 48.43(0.51) | 50.86(0.19) | 51.01(0.31) |
| 28 | 44.51(0.61) | 49.55(0.59) | 52.23(0.45) | 52.16(0.32) |
| 32 | 46.52(0.86) | 50.68(0.38) | 53.21(0.34) | 53.11(0.22) |

Table S.3: Mean average precision of nearest neighbours search on 10000 random ordered samples in MNIST. The numbers are in percentage and the numbers in brackets are variances.

| $k$ | FlyLSH | Method 1 | Method 2 | Method 3 |
|---|---|---|---|---|
| 2 | 3.43(0.32) | 6.88(0.35) | 7.56(0.16) | 7.48(0.37) |
| 4 | 7.39(0.55) | 13.04(0.43) | 14.33(0.24) | 14.22(0.43) |
| 8 | 15.72(0.60) | 21.78(0.48) | 24.31(0.37) | 24.34(0.31) |
| 12 | 22.86(0.52) | 27.51(0.59) | 30.84(0.28) | 30.82(0.36) |
| 16 | 27.93(0.37) | 31.25(0.39) | 35.48(0.33) | 35.10(0.20) |
| 20 | 32.02(0.64) | 34.07(0.67) | 38.28(0.33) | 37.93(0.41) |
| 24 | 35.55(0.70) | 36.13(0.65) | 40.14(0.40) | 40.26(0.43) |
| 28 | 38.33(0.70) | 37.80(0.40) | 42.21(0.23) | 42.04(0.36) |
| 32 | 40.26(0.68) | 39.00(0.59) | 43.23(0.36) | 43.47(0.31) |

Table S.4: Mean average precision of nearest neighbours search on 50000 random ordered samples in MNIST. The numbers are in percentage and the numbers in brackets are variances.

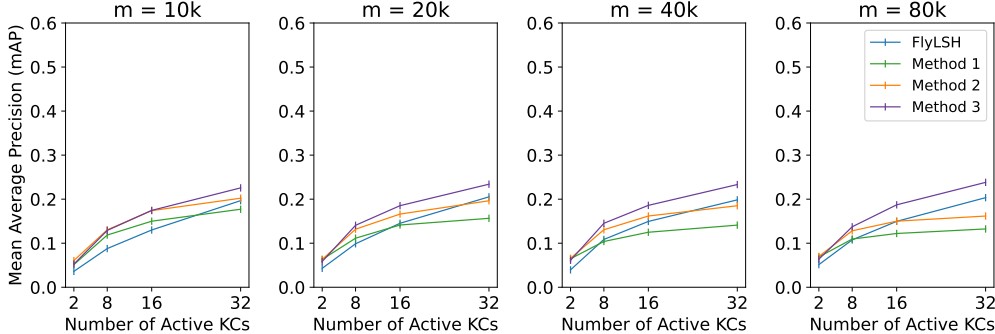

Figure S.1: Performance on CIFAR-10, size of the dataset: 10000, 200 nearest neighbours.

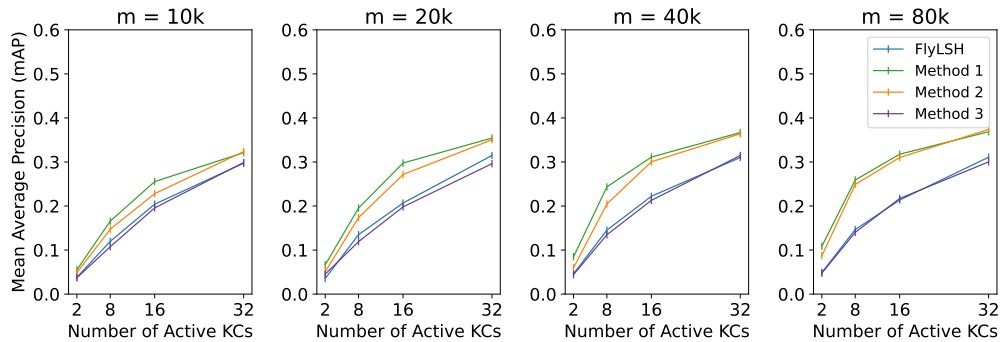

Figure S.2: Performance on GloVe, size of the dataset: 10000, 200 nearest neighbours.

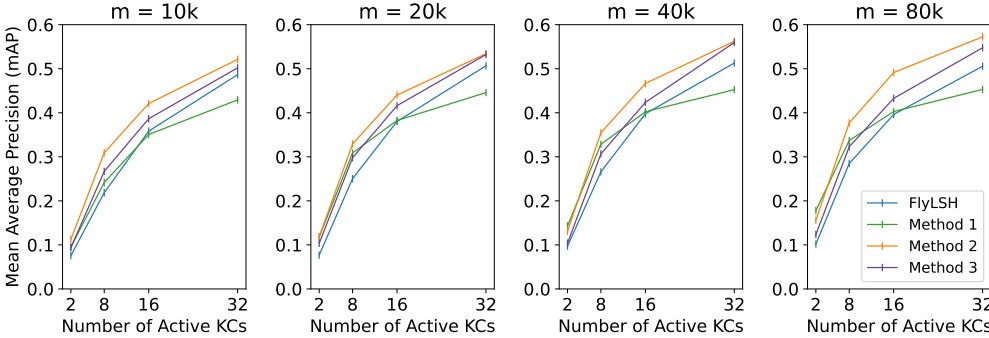

Figure S.3: Performance on SIFT10M, size of the dataset: 10000, 200 nearest neighbours.

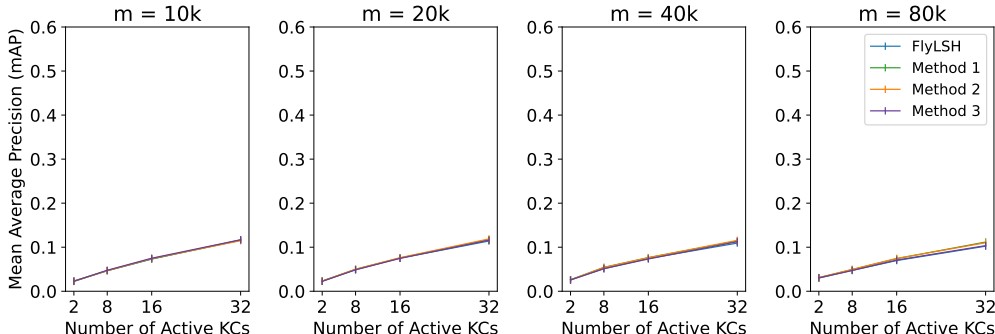

Figure S.4: Performance on RANDOM, size of the dataset: 10000, 200 nearest neighbours.

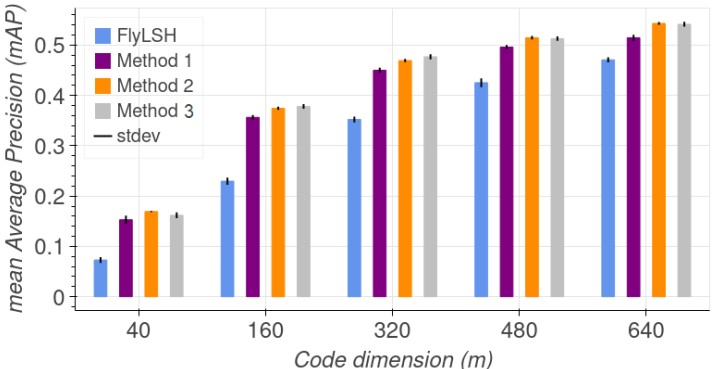

Figure S.5: Nearest neighbours search after training on ordered samples in MNIST.

## S.B.2 Receptive fields

One way to interpret the PN-KC connections is to consider each KC to have a 'receptive field', i.e. an area in the image that produces the strongest response. We can visualise the effective receptive field produced by FlyLSH (Figure S.6A left) or DevFLY (Figure S.6A middle) with the latter clearly reflecting the data structure. For interest we compared the performance if we used pre-defined spatial receptive fields. That is, for each KC, we chose a random mean and radius for a 2D Gaussian distribution and made $n$ connections from this region of the input to a KC. Examples are visualised in Figure S.6 A, right. However, as shown in Figure S.6 B, the performance was worse than random connections (FlyLSH) on MNIST and so we did not pursue this approach further.

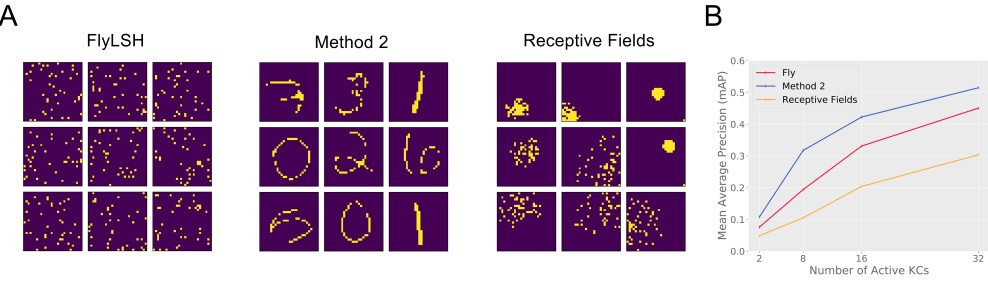

Figure S.6: (A) Examples of the weights for 9 KCs in FlyLSH, Method 2, and the receptive fields method. (B) Performance of these 3 methods on MNIST using $\alpha = 0.05$ and $m = 20k$.

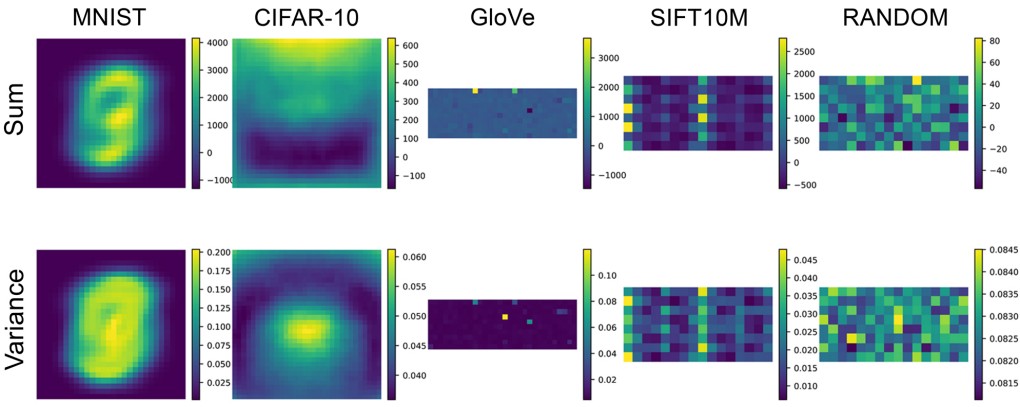

Figure S.7: Sum of the pixel values across the samples (above), and variance per dimension (below).

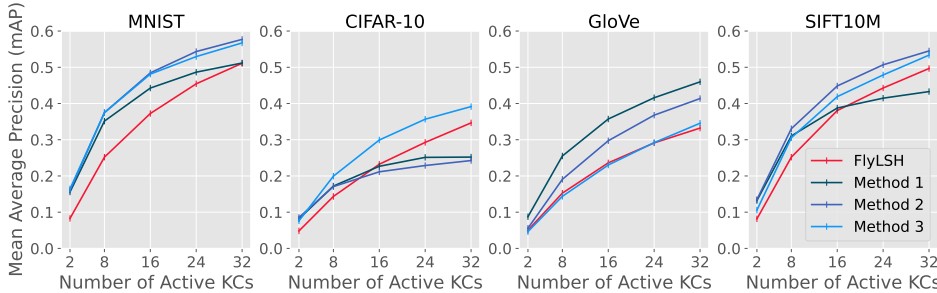

Figure S.8: Performance with normalised datasets. 10000 samples were taken from each dataset for the tests. 1000 queries for 200 nearest neighbours were made for the means and variances. $m = 20k$, $\alpha = 0.1$.

### S.B.3 Performance with random data

One solution for this could be to implement neural adaptation as a pre-processing step, such as high-pass filter, and use the pre-processed signals to determine the PN activation. The pre-processing happens in the insect brain. Sensory inputs are pre-processed in different insect brain regions, such as the antennal lobe or the optic lobe, before being fed to the MB (Li et al., 2020), and the optic lobe can high-pass filter the visual inputs (Borst, 2009). A second simpler option could be to just use the difference from one sample to the next to determine the connections, that is, the pixels that change the most between successive samples are chosen to connect to the new KC, rather than those with highest intensity in the current sample. A third option might be some pre-processing of the data such that pixels with low variability are detected and made ineligible for connecting to KCs. It might be sufficient to use a small subset of the data for this pre-processing step.

### S.B.4 Performance with normalised data

Because of the winner-take-all process in the binarization of the hash code, the information of the samples' $l^2$ norms is lost. Hence, if samples in a dataset have the same $l^2$ norms, which means the samples located on a hypersphere, FlyLSH and DevLSH should perform better. It is supported by our experiment on normalised datasets, by which their $l^2$ norms are the same. The samples in the normalised datasets were hashed and queried, then the mAP was measured. Figure S.8 shows the result. Comparing to Figure 4, the performance with normalised CIFAR-10 is better than the performance with the original dataset.