# OpenReview forum: "DevFly: Bio-Inspired Development of Binary Connections for Locality Preserving Sparse Codes"
_NeurIPS.cc/2022/Conference — NeurIPS 2022 Accept_

### Official Review · Reviewer_oFUq · 2022-07-11

**Rating:** 7
**Confidence:** 5
**Soundness:** 4 excellent
**Presentation:** 4 excellent
**Contribution:** 3 good

**Summary:**

This paper proposes a new locally-sensitive hashing scheme based on the FlyLSH – a prior work where the hashes were obtained using a neural network modeled after the fruit fly’s olfactory system. Whereas in the FlyLSH the network’s weights were random, here the authors propose to connect the strongest input dimensions to distinct output neurons for distinct stimuli. The authors test their model on a standard set of tasks (MNIST/CIFAR/GloVe/SIFT) where they report an increase in performance.

**Questions:**

No questions at this time.

**Limitations:**

The limitations are discussed adequately.

**Strengths And Weaknesses:**

This work presents an important and timely contribution to the field of locally-sensitive hashing. After the introduction of the FlyLSH, where the network weights are random, the next logical step was to optimize these weights in a data-driven manner to achieve better performance. This work offers an elegant and computationally light way to meet this goal.

Among the strengths of this work are:

-The computational simplicity of the proposed method. The learning of the weights can be performed online as the new samples, substantially different from the already-learned ones, are assigned the new KCs on the output layer. The evaluation is also fast due to the fixed-length sparse code which allows finding the nearest neighbors in a small number of comparisons.

-The thorough testing of the method. The authors have tested their proposed model on four standard benchmarks in a way compatible with analogous testing in previous works; they show an increase in the model’s performance compared to the FlyLSH baseline. Besides, the differences in performance are discussed for the data types in datasets and the best use cases for the model are defined.

-The text is well-written and well-structured including a comprehensive introduction and a thorough description of the related work.

-The implications of this work for neuroscience may be interesting. Intuitively, the model may replicate: i) the difference in the projection densities of the ORs, ii) the lack of the “grandparent cells” among the KCs, i.e. the cells encoding individual odors, and iii) the “primacy” hypothesis stating that the odors may be encoded by the sets of the earliest-responding glomeruli.

For these reasons the paper appears to be a substantial contribution relevant to the NeurIPS community

---

> ### Author Response · Authors · 2022-08-02
> **Thank you very much for the comments and affirmation**
>
> Thank you very much for the comments and affirmation that this paper makes an important and timely contribution.

---

> > ### Comment · Reviewer_oFUq · 2022-08-03
> > **Re: response**
> >
> > My pleasure.

---

### Official Review · Reviewer_8FqY · 2022-07-11

**Rating:** 4
**Confidence:** 3
**Soundness:** 3 good
**Presentation:** 3 good
**Contribution:** 2 fair

**Summary:**

This paper suggests a new architecture for nearest neighbor search. The paper uses a new algorithm to design
a neural network that solves NN that is inspired by circuitry in the insect brain. In contrast to previous approaches that use random connections  the paper suggests a non random architecture. The new method outperforms previous neuro-inspired methods in several metrics such as search time and accuracy.

**Questions:**

I would elaborate more on the primary goal of this paper. Also some mathematical proofs of what the algorithms achieves can strengthen the contribution.

**Limitations:**

In contrast to Dasgupta, S., Stevens, C. F., & Navlakha, S. (2017) no provable guarantees are given regarding the performance of the algorithm.

**Strengths And Weaknesses:**

The experimental results are nice and the algorithms is interesting. I was uncertain about the end goal of this paper.
Is it to find the best network for NN? If yes much more comprehensive comparisons should be made. Is it to give new explanations as to how the insect brain solves NN? That connections are typically non random in biological neural networks is long known. E.g.,
"Highly Nonrandom Features of Synaptic Connectivity in Local Cortical Circuits"

---

> ### Author Response · Authors · 2022-08-02
> **We stated the goal of the paper in the abstract**
>
>
> Thank you for the comments.
>
> This work is one of our efforts to build a mutually beneficial connection between neuroscience and machine learning, with a methodology that biologically plausible hypothetical principles can be tested on some machine learning tasks for a preliminary evaluation, and if the evaluation result is encouraging, the model could contribute to both sides.
>
> We stated the goal of the paper in the abstract "We investigate whether the performance of the hash, evaluated in nearest neighbour query tasks, can be improved by [an MB-inspired] process of developing the connections...".
>
> More specifically, our goal is to define a new developmental rule that can be used to improve the performance of FlyLSH while preserving its two requirements: sparse connectivity and sparse hash code. Since the Mushroom Body fulfils both of these requirements, it is a good place to look for inspiration. Note that we are only claiming that this method is better considering the constraints we are working with, so it is not fair to compare it to other methods in the field that don't take into consideration these constraints. Since the main goal of LSH is to speed up both the computation and the search for nearest neighbours in high dimensional datasets, a sparse connectivity matrix is important to speed up the hashing, while a sparse hash is needed to speed up the nearest neighbours query.
>
> If the reviewer thinks it would be helpful we can restate this at the end of the introduction in a revised version.
>
> We agree that mathematical proofs would strengthen the contribution and are working on these to hopefully include in the final version.

---

### Official Review · Reviewer_Dki9 · 2022-07-14

**Rating:** 6
**Confidence:** 4
**Soundness:** 3 good
**Presentation:** 3 good
**Contribution:** 3 good

**Summary:**

Neural circuits present in the fly mushroom body provide a nice biological analog of a simple feedforward artificial neural network. Such circuits also exist in cerebellum and have been studied both from the lens of supervised learning and unsupervised representations. This paper looks at the unsupervised aspect, studying how such networks can construct locality-sensitive embeddings/hashes into higher dimensions. The main contribution of this work is to study data-dependent ways for these circuits to embed. Three methods are presented and compared to some past methods. These data-dependent hashing methods are shown to offer improvements in some situations, and a few hypotheses as to when/why these methods do or do not work better are presented.

**Questions:**

Can you provide code?

Algorithm 1/2: The notation needs to be explained. What is $\mathbf{i}'$ in particular its dimension? How do you determine the "top n values", by what metric? What range can $\beta$ cover? What is $\mathbf{y}'$ initialized to? What does it mean for $\mathbf{y}' > \theta$ if $\mathbf{y}'$ is a vector? What is the variable $u$? Unless these issues are all addressed, the algorithms are unreproducible.

Last paragraph of Section 4.2: The claim that CIFAR-10 is different due to differences in magnitudes suggests that standardization would fix this problem. I don't see how CIFAR-10 isn't clustered, there are 10 classes, which is stated in line 338. Not much evidence is presented for the claim (272) "LSH methods work best on datasets that are clustered and samples that lie on the hypersphere".... No clear definition of clustering is given. Also, sphereing the data by normalization or standardization is a simple thing to try to support your claim.



**Limitations:**

There needs to be more discussion of limitations of the methods proposed. Also, the support for many of the claims as to why the methods work in some cases and not in others is limited, and the authors shouldn't hide this.

The discussion of societal impact is almost non-existant and seems an afterthough. However, the authors mention "security camera video" as a potential data source (344). What kind of ethical issues could come with this? I suggest doing more here.

**Strengths And Weaknesses:**

Strengths: The paper is overall pretty well-written and the methods are tested on multiple datasets. The motivation from neuroscience and computer science fits well into the (historical) focus of NeurIPS.

Weaknesses: The main weakness is that code is not provided and the algorithm descriptions in the paper are confusing, especially the notation that was used. I do not think I could reproduce the paper's results without code. It is unclear whether the comparison of runtimes with other methods is entirely fair because the implementation details are not fully-described. Some claims of why some methods work or do not (in terms of "clustered" data) are not well-justified.

Specific critiques by line number:
36: parenthetical "(if the hash length...)" is awkward, rephrase

Fig 2 caption: NHL, ALH, APF abbreviations don't get used elswhere, could remove to make figure more readable

53: should cite Sophie Caron's latest work beside Eichler and Elkahlah

54: Litwin-Kumar et al studied random connections, so this theoretical study is not a good one to refer to here

90: corpora should be "corpus"

111: "and this form of network can be related to radial basis functions" is unclear

121: The summary of FlyLSH in this section is long and not the method which is presented. I recomment shortening this to allow a better description of the novel methods.

133: "should not be sensitive to this..." this claim is unclear, because you haven't presented your own methods yet.

134: "most intensive pixels" unclear, why should we be thinking about pixels? Any kind of data could be used.

143: use \leq instead of <=

144: use \ll instead of <<

155: You need to spend much more space describing the novel methods. This is the most confusing part of the paper.

Algorithm 2 line 4: I believe you lack and "and" between $\mathbf{y}' > \theta$ and $\exists$.

198-199: This should be earlier in Section 3.

200: I found the description of the datasets to be very good.

213: 1000 to 1,000 (be consistent throughout document, there are multiple instances of this)

214: Suggested rewording "For each query sample,"

215: capitalize "Hamming", strikeout "sample,"

217: remove ", that is,"

225: This paragraph seems unnecessary, out-of-place, and potentially confusing. I would put it elsewhere.

234: WTA not defined

245: mean average precision (mAP) not defined

Fig 3: It is unclear how the algorithms were implemented and how this might influence the runtimes

Fig 5: Define "magnitude variance"

285: change "MB inspired" to "MB-inspired"

Fig 6: use curly braces 10^{-2}

299: Method 1 doesn't depend on the original data order, so this isn't surprising.

302-304: "The threshold..." sentence needs more explanation

322: "... properties of the data" would benefit from supporting citations

---

> ### Author Response · Authors · 2022-08-02
> **The code was provided and please see answers to other questions**
>
> Thank you for the comments on our paper. The suggestions to revise by line number are very helpful and we would adopt all these suggestions. Please see the rebuttal revision.
>
> # Answer to question 1
> The code has been provided for review since the first version. It was in the supplementary material we submitted, packed in a zip file named DevFly.
>
> # Answer to question 2
> 1. $\mathbf{i}^\prime \in \mathbb{Z}^n$ is a set of indexes of a sample's dimensions.
> 2. The "top n values" are determined by sorting the sample dimensions by their input value and taking the n largest.
> 3. $ \beta $ is a value that is slightly larger than 1. In practice, we used $\beta = 1.2$.
> 4. $ \mathbf{y}^\prime $ is a typo and should be $y^\prime$.
> 5. ${y}^\prime$ is the computational result of the product of a KC's weights and a sample, as shown in Algorithm 2 line 6. We realise a clearer way to express the constraint ${y}^\prime>\theta$ would be:  WHILE $ \exists$  $j^\prime \in \{1,2, \dots, j \} $ such that $\mathbf{r}_{j^\prime} \mathbf{s}_q > \theta $
> 7. $u$ is the number of data samples.
>
> These notations are updated in revision.
>
> #  Answer to question 3
> "Clustered" here refers to the distribution in space of the original dataset, and the degree to which this distribution is clumped rather than evenly spaced. We did not claim that CIFAR-10 isn't clustered (On line 272 we say: "However, for CIFAR-10 the problem appears not to be lack of clustering..."). Our point is that CIFAR-10 samples (within a class) have a larger variance in magnitude (or $l_2$ norm) than samples in MNIST. As the reviewer suggested, we have tested the result of normalising the norm of samples in CIFAR-10 to lie on the hypersphere and this does improve the results (we can include these in an appendix) but we note that this does not solve the original problem as it distorts the "nearest neighbour" property in the original data.
>
> We would clarify these in the revised paper by replacing the paragraph from line 266 as follows:
>
> "Higher performance of the models on MNIST and SIFT10M than on GloVe is, we believe, due to the data in MNIST and SIFT10M lying on a lower dimensional manifold, relative to their input dimensions, than GloVE, and hence requiring fewer samples to adequately represent the space. Principal components of the variables of a dataset can give us a good indication of the size of this space, and as shown in figure 5A, the cumulative sum of explained variance by principle components increases much more slowly for GloVe than the other datasets. This is confirmed by observing that none of the models was able to find more than 10\% of the nearest neighbours when tested on a randomly generated dataset.
> For CIFAR-10, instead, we observe that, ideally, to preserve the nearest neighbours in the input space, each KC (which represents a centroid in space) should be activated proportionally to the Euclidean distance between the input sample and the centroid. However, as the intermediate code is actually defined by the dot product, it is scaled linearly in the direction of the vector represented by the weight. This means that KCs are not activated locally but are biased by samples that have a bigger norm.
> This is not the case with datasets that have samples with equal l2 norms, as the dot product is local when samples lie on the same hypersphere (SIFT10M) or have approximately this property (MNIST). Experimentally, we can test this by checking if equalizing the norms improves the performance, and we indeed see a significant improvement for CIFAR-10. However, if the goal is LSH, equalizing the norms should not be considered to be an acceptable pre-processing step, as it does not preserve the distances of the original dataset."
>
> # Answer to limitation 1
> As in our answer to question 3, we will revise our discussion of why the methods work in some cases and not others to make this better supported, and hopefully this also clarifies some limitations of the method. We would also add to a revised paper this sentence in the discussion section, at the paragraph starting line 321 (comparing to other methods for optimising connections to preserve locality): "A limitation is that this process is not guaranteed to produce the optimal solution, but as we have shown it can nevertheless be effective for LSH problems".
>
> # Answer to limitation 2
> In a revision we would briefly expand on the societal impact as follows: "LSH nearest neighbour classification can be used for a very wide range of applications from security camera recognition to medical image analysis. By introducing a biologically inspired method we might hope that the resulting system would more closely resemble the decisions made by humans and hence have a positive societal impact. We also note that the simplicity of the approach makes it suitable for edge computing, which could allow data (such as that from security cameras) to be processed locally, with consequence privacy advantages in limiting data transfer".

---

> > ### Comment · Reviewer_Dki9 · 2022-08-08
> > **Thanks for addressing limitations**
> >
> > I hope that the final version takes this all into account and is clearer. I've revised my scores upwards.

---

### Meta-Review · Area_Chair_SvEK · 2022-08-24

**Recommendation:** Accept
**Confidence:** Certain

**Metareview:**

This paper provides learning algorithm to learn connections in a biologically-motivated network for locality sensitive hashing. This contribution extends previous work where such connections were randomly chosen. I recommend acceptance based on the overall sentiment of the reviews.

**Award:**

No

---

### Decision · Program_Chairs · 2022-09-14

Accept